# Elements of Designing Upholstered Furniture Sandwich Frames Using Finite Element Method

**DOI:** 10.3390/ma15176084

**Published:** 2022-09-02

**Authors:** Łukasz Matwiej, Marek Wieruszewski, Krzysztof Wiaderek, Bartosz Pałubicki

**Affiliations:** Department of Mechanical Wood Technology, Poznan University of Life Sciences, Wojska Polskiego 28, 60-627 Poznan, Poland

**Keywords:** layered stile of upholstered furniture, element connections, stresses, deformations, FEM

## Abstract

This paper presents an approach to the design of an upholstered furniture frame using the finite element method and empirical studies. Three-dimensional discrete models of upholstered furniture frames were developed taking into account orthotropic properties of solid pine wood (*Pinus sylvestris* L.) without and with details strengthening their structure in the form of glue joints and upholstery staples. Using the CAE Autodesk Inventor Nastran finite element method, linear static analyses were performed by simulating normative loading. The finite element method was performed considering the experimentally determined stiffness coefficients of the PCAC adhesive and staple joints. As a result, stress, displacement, and equivalent strain distributions were obtained for upholstered furniture frame models with stapled corner joints. The deformation and strength behavior of the upholstered furniture frames was improved by reinforcing with a wood strip. A new approach to the design of upholstered furniture frame frames using the FEM method with stapled component connections was developed and tested. The results of the study can be applied in the optimization of upholstered furniture construction.

## 1. Introduction

Wood furniture components are examples of structural elements used in both box and frame furniture systems [1,2]. Upholstered furniture frames are often structural subassemblies made of several to as many as a dozen boards joined together to form cohesive structural elements. They are often referred to as furniture stiles. Stiles can be of solid or layered construction, where the strips of lumber used can be joined in parallel or crosswise (known as glued stiles). The wood material used in the construction of upholstered furniture is usually wood with a certain strength and susceptibility to processing. In the group of European raw material, coniferous species, such as spruce and pine, and deciduous species—such as birch and beech—play a dominant role [3,4]. They meet the strength requirements and account for a significant percentage of the renewable raw material available for processing. In this group, pine and spruce wood deserve special attention. These species, in addition to their high availability, have characteristics that allow them to be used in lightweight furniture structures [3,4]. Natural and modified wood materials simultaneously have strength parameters that meet the requirements for furniture structure design [5,6,7,8].

An important feature of bonded stiles, also known as frame subassemblies, is that the elements (stiles) in frame nodes are connected in such a way that mainly lateral forces act in them. To ensure correct stress distribution, the external load must be applied directly to the stile plane. The cross-sections of wooden elements are much smaller than their length. The effect of dead weight of the stile structure on the level of internal forces is negligible in such furniture solutions. In a traditional situation, it can be assumed that there are sets of shear forces and bending moments in the framing elements from the weight of external load. The relatively simple assumptions used to determine the stresses occurring in the frame elements, i.e., stile elements, allow the use of various techniques and algorithms to optimize their dimensions and shape [9,10]. The advantages of wooden structures are related not only to the simplicity of their manufacture, but also to their durability, light weight, ease of shape modification, and other positive properties of wood [11,12,13,14,15]. Another favorable feature of wooden structures is related to the reduction of carbon footprint, which has a significant impact on the environment, especially in light of the concept of sustainable development [3,4].

In the case of wooden frames, the high ratio of stiffness and load-bearing capacity to the amount of material used is also emphasized. Stiles can be manufactured both as small-size and large-span components. The positive aspects of joining wood slats or wood can be eliminated as a result of improper connection of stile elements.

The joint must be designed to carry the assumed load without losing the rigidity of the system [9,16]. As mentioned above, this can be achieved relatively easily with adhesive or steel joints in the form of staples or screws [17,18]. Properly selected staples driven into the layered bonded wood provide greater strength and durability to the framing elements, as the placement and number of staples provide an adequate attaching surface [19,20,21,22,23]. Upholstered furniture stiles must be strong and rigid enough to withstand static and dynamic loads during use. The strength and deformation characteristics of upholstered frame elements are very important to ensure optimal construction of upholstered furniture.

Nowadays, strength design of furniture structures can be carried out using 3D modeling and structural analysis software based on the finite element method (FEM). The popularization of numerical methods and the availability of CAD 2010–2019 software have facilitated the design process of wood-based systems [24,25,26,27,28]. Presentation of the principles of furniture design as wooden components of furniture construction, with classification and their characteristics, is concerned with taking into account the requirements of safety in use. Basic strength testing methods can be prone to design errors, characterization of materials, elements, and structures. The FEM method addresses the issue of calculating and verifying the stiffness and strength of components, joints, and entire structures, including issues of user health hazards. The use of modeling for the evaluation of furniture structures is to serve as one of the primary sources of knowledge for the formulation of design assumptions for a new product, which often reveals deviations from the general assumptions of structural design, which significantly improves the process of finding solutions that satisfy the creators of a new piece of furniture. The FEM method is a developmental one for the direction of designing furniture as complex structures. The FEM method is an engineering tool to help solve technical problems in furniture design.

Eckelmann and Suddarth were the first to propose the application of numerical methods to furniture design, preparing a special program package in Fortran IV-CODOFF and CODOC-3 [29]. Some of the first numerical studies of upholstered furniture frame structures were undertaken by Altınok and Örs [30] or Smardzewski [31]. Initially, a well-known finite element analysis [32,33,34,35] of the furniture frame was conducted to reduce material consumption and ensure optimal structural strength using the CAE algorithm. However, an unjustified modeling of wood and particleboard elements as isotropic materials was proposed. The discrete model does not take into account the actual behavior of structural element connections of the furniture connectors. In the modulations carried out, a CAE algorithm was used to optimize the dimensions of the main structural elements of frame modeled with basic orthotropic finite elements (including pine and beech wood) focusing on the finite element network where elements of different thicknesses and cross-sections connect, in order to obtain an accurate image of the deformation and stress state [36,37]. Kasal [38] investigated the strength properties of glue-bonded furniture frames made of solid wood and wood-based composite materials using RISA 3D finite element analysis software 2010–2020. Taking the wood materials as isotropic, he found that OSB had the lowest load capacity. Wang [39] conducted a nonlinear static analysis using SAP2000 finite element analysis software 2000 of three configurations of a furniture frame made entirely of OSB. The product was modeled using beam elements with two types of connections (screws with metal plates and staples with metal plates). Wang [39] used rigid and semi-rigid connection types in the models, but introduced the experimentally-determined linear-elastic stiffness of connections rather than rotational stiffness into the program. Researchers [40,41] modeled wood furniture frames with isotropic characteristics using ANSYS Workbench software 2. The results confirmed that there was convergence between experimental and FEM results (81% level of agreement). The study of wooden chair frames using the finite element methods was undertaken by researchers [42] using the stress-strain methodology developed by Marinova [43] for FEM-based analysis of box furniture structures. Tests were adapted to the characteristics of box furniture structures and the elastic constants determined in tests were imported into the SAP 90 computer program.

The aim of this paper was to verify a new approach to the design of a layered stile using the CAE system Autodesk Inventor Nastran, which is an element of the upholstered furniture frame, taking into account the experimentally determined stiffness coefficients of the applied elements using glue joints. The solution proposed in the publication in the form of a system of evaluation and selection of materials—after verification in further analyses—should facilitate the process of design and rational management of available wood resources.

## 2. Materials and Methods

Stiffness and strength analysis of selected stile structures in the form of 3D models was carried out virtually in Autodesk Inventor Nastran version 2020 using the finite element method (FEM). This method involves abandoning a continuous model of the structure in favor of dividing it into a finite number of elements, called finite elements [44,45,46]. The distribution of reduced stresses as well as the distribution of deformations was obtained by using discrete models and an advanced iterative method defined in the program’s solver. FEM is a tool designed to solve differential equations, and more specifically to find detailed solutions. The discretization of the structure of individual frames was based on the automatic division of all elements of the model into appropriately selected finite elements of defined shape and properties, connected to each other at specific points called nodes. All models were automatically divided into nonlinear finite elements with an assumed element size of 10 mm. The models were loaded exactly as they were in experimental tests on real elements. This made it possible to automatically visualize the distribution of stresses and equivalent strains.

The study separated four raw material groups based on the origin of raw material in the selection of pine wood labeled PNSY (*Pinus sylvestris* L.) from Poland (PL) [47], Eastern Europe (EW), and Scandinavia (SC) [48,49] and spruce wood labeled PCAB (Picea abies) from Scandinavia (SC). The characteristics of wood raw material are related to the strength properties that characterize the behavior of materials in the modeling process, such as elastic modulus, strength, required in the manufacture of furniture products [50,51,52,53]. As an orotropic material, wood has variable mechanical properties depending on the considered direction of loads. In engineering terms, this means that it is necessary to take into account the directions of forces s in relation to the direction of the wood fibers. It transmits stresses along the fibers much better than across them. The use of wood glued longitudinally and lengthwise with a finger joint (a 13 mm finger-length furniture joint using PVAC glue) and on the one hand reduces the influence of the orotropic features of the tangential and transverse sections but does not eliminate the influence of the longitudinal section. However, for the purpose of FEM modeling of force distribution in the normative loading process, the reference to the stresses acting in the longitudinal structure is mainly considered. This allows a partial simplification of the model of the glulam component under study.

Test of stiles bending strength

*a.* 
*Construction of 3D models of stiles*
-Before working in Autodesk Inventor Nastran program on the basis of engineering sketches using Autodesk Inventor, continuous geometric models were developed, i.e., 3D models of the stile structure. Due to the adopted principle of maximum simplification of the geometry of continuous models, dictated by the need to minimize the number of finite elements, the joints (forks) were not shaped in stile models, and also the model was simplified by not introducing the glue joint as an additional element. The elements to be tested are a long frame board, a short frame board, a long frame dovetail, a long frame short dovetail and a short frame dovetail. At the same time, the frame boards have a cross-section of 20 mm × 80 mm, while their dovetails are 20 mm × 48 mm. Due to the adopted research plan, a total of 6 stile models were prepared:

A—long made of Solid wood material,

B—long made of glued material with a finger-jointed lamella length of 200 mm,

C—long made of glued material with a finger-jointed lamella length of 400 mm,

D—short made of solid wood material,

E—short made of glued material with a finger-jointed lamella length of 200 mm,

F—short made of glued material with a finger-jointed lamella length of 400 mm.

Due to the need to realistically reflect the experimental (laboratory) tests conducted in parallel, supports and thrusters were introduced into each 3D model to simulate four-point bending tests with the following spacings:-for long stile, support spacing 1960 mm, thrust spacing 600 mm, thrust distance from the support 680 mm,-for short stile, support spacing 810 mm, thrust spacing 270 mm, thrust distance from the support 270 mm.

An example of 3D models is shown in Figure 1, Figure 2 and Figure 3.

*b.* 
*Idealization of design*


Obtaining glued laminated timber stiles with appropriate parameters is related to meeting the minimum strength requirements according to the subject standard for furniture products in accordance with PN-EN 1725 [54] classifying semi-finished products in accordance with PN-EN 338 [55] according to the methodology of PN-EN 14081 [56], which is the level of section strength to values in the range of 24 N/mm^2^ and 30 N/mm^2^. The assumed load on the stiles was assumed to be 2400 N in the static vertical load test of t stiles, by applying two forces of 1200 N at a time. The forces should be applied so that they are 600 mm apart and 300 mm from the center of stile.

Modeling studies and laboratory tests are based on strength-quality limits that allow to achieve the required quality by selecting the characteristics of selected semi-finished products or stiles. In case of modeling of the strength of long stiles for the A-C frame, the selection of solid pine raw material classified according to the assumptions of visual standards PN 94021:2013 [57] was made, which, according to the appendix of PN-EN 338 [55], were assumed to be in accordance with the division into classes KW—corresponding to C30: the strength of 30 N/mm^2^, KS—corresponding to C24: 24 N/mm^2^ [6,55,58,59]. These studies were supported by the verification of defects in the form of threading, knots, fiber twist, or multilaminar connection errors for semi-finished length-jointed stiles, shown as the basic elements to be evaluated in the optimization process of modeling [60,61]. For D-F short stiles, the model values reached the level of KS—corresponding to C24: 24 N/mm^2^ and KG—corresponding to C16:16 N/mm^2^ strength [55].

The assumed requirements are met by spruce and pine raw material from Scandinavia, Poland, and Eastern Europe. The limit values were established based on laboratory tests of the experimental raw material. The test method used for evaluating laboratory results and output for subsequent modeling is in accordance with EN 408 [62]. The system of strength classes is based in Europe on the provisions of standards, such as EN 1912 [63] or EN 338 [55]. In the study, it was assumed that, despite the existence of differences in the tested parameters for machine strength testing and visual evaluation 94,021 [57] of solid elements, it is possible to take into account the results of laboratory self-testing d modeling of the construction of glued frames.

Analysis of stiffness and strength of selected frame structures, in the form of 3D models, was carried out virtually in Autodesk Inventor Nastran version 2020 using the finite element method (FEM). Continuous geometric models were developed, i.e., 3D models of the frame structures. Due to the need to reflect experimental testing, supports and thrusters were introduced into each 3D model to simulate a four-point bending test. Next, the so-called idealization was carried out, which involved assigning specific physical properties of the raw material (material) to the individual frame members. The physical properties of each type of raw material were derived from parallel experimental (laboratory) tests. The next step was to automatically divide all model elements into appropriately selected finite elements of defined shape and properties, define boundary conditions in the form of determining the number and type of degrees of freedom at selected nodes, assigning interactions (contacts) between selected surfaces, and introducing external loading of the structure. After preparing the models for calculation, numerical analyses were carried out in the linear static analysis mode. It should be noted that the authors made a number of tests and comparisons of the results obtained through experimental studies and numerical analyses before conducting the actual numerical analyses.

Idealization of design was based on the assignment of specific physical properties of the raw material to the individual elements of stiles. The physical properties of each type of raw material were derived from parallel experimental (laboratory) studies. The exact properties of materials used are shown in Table 1, Table 2, Table 3 and Table 4. Dimensionally, the experimental material corresponded to the assumptions used in the modeling process. Laboratory results are presented as averages of each series (50 samples).

*c.* 
*Structure discretization*


There are two methods to achieve the discrete model and one of them, used during the analysis in this paper, is FEA—finite element method [64]. The FEM method is an approximate method [38,40,64]. It is also a numerical tool for solving differential equations, and more precisely to find detailed solutions [65].

Discretization of individual stile structures consisted in automatic division of all model elements into appropriately selected finite elements of defined shape and properties, connected to each other at specific points called nodes. All models were automatically divided into nonlinear finite elements (parabolic type) with an assumed element size of 10 mm. The appearance of selected mesh (computational) model is shown in Figure 4.

*d.* 
*Introduction of boundary conditions*


The next step was to define boundary conditions in the form of specifying the number and type of freedom degrees at selected nodes, assigning interactions (contacts) between selected surfaces, thus simulating adhesive connections of elements or surface-to-surface interactions, and introducing an external load on the structure.

The following boundary conditions were introduced in all 6 stile models:-both supports were deprived of all freedom degrees (all translations and rotations—the so-called restraint),-both thrusts were deprived of their ability to rotate and also slide across the stile,-in order to achieve convergence (convergence of obtained calculation results) in one contact between the element support and surface, a friction coefficient of 0.5 (static friction coefficient for the steel-wood contact pair) was introduced,-an external load was applied to the thrust upper surface in the form of a force normal to the surface directed vertically downward with a value of F = 1200 N (total load 2400 N),-interactions of contacting stile surfaces with supports and thrusts were introduced in the contacting surfaces on the principle of separation-type surface contact,-interactions were introduced in the contacting surfaces of stile elements on the principle of bonding (without adhesive-bonded joint).
*e.* *Assigning numerical analysis settings*


The numerical analysis was carried out in static linear analysis mode.

Stiffness of a given structure (k) is measured as the ratio of force acting on the product which is an external load (*Pz*), to the displacement measured in the direction of this load (∆*Pz*):$$k=\frac{Pz}{\Delta Pz}\frac{N}{m}$$

The aim of the conducted research was to analyze the stiffness and strength of stile structure by determining the values of element displacements in the direction of external load. In the material area, the evaluation of coniferous wood raw material from European areas with the nature of renewable materials with a wide range of strength characteristics was separated for the study.

## 3. Results and Discussion

After conducting numerical analyses of the modeled furniture frames, the results were obtained as follows:-maps of the displacement distribution of stile structural elements, counteracting slats, edge of bed, and also the bed frame on the direction of load,-maps of the normal and or reduced stresses distribution developed in the stile structural elements, counteracting slats, edge of bed, and also the bed frame.

The analyses carried out showed that the proposed method of loading the structure and, consequently, deformation of stile elements results in stresses similar in value in all analyzed stile variants of a given group regardless of the wood raw material to be used. If these stresses were close to the strength limit of a given material, they could threaten the strength of a given element and thus the load-bearing capacity of the stile structure. Conducted load simulations showed that at the critical points of structure (the vicinity of the maximum deflection arrow) the normal stresses do not exceed the bending strength limits of the material. For all considered stile models, normal stresses in the upper flange show negative values, while in the lower flange they show positive values. Thus, in a generalized overall view, the stiles behave like a classical beam, in which during bending it is possible to observe a compression zone, a tensile zone and a rather narrow stress-free strip—the so-called bending neutral zone. Figure 5, Figure 6 and Figure 7 show examples of normal stress distribution maps for selected stile models.

A full summary of the calculated values of normal stresses as well as displacement (deflection) in the direction of external loading for all analyzed stile models using several types of raw material is listed in Table 5, Table 6, Table 7, Table 8, Table 9 and Table 10.

In the case of the analyses given, all modeling was carried out with the same constant load. Thus, in simplification, it can be assumed that a direct measure of stiffness in this case is the displacement (deflection) of the loaded element measured in the direction of load. It was found that in the tests conducted, the stiffness of a given structure decreases as the value of deflection shown increases. Analyzing the stiffness of all long stiles, it should be concluded that in the case of both solid and glued structures, the most matched element to the requirements of minimizing deformation in most cases is a stile made of pine wood from Scandinavia, and the least matched (the largest deflection arrow of element) is a stile made of pine wood from Eastern Europe. The maximum differences in the obtained values of deformation reach about 22%. Similar dependencies of the adjustment of wood type to the stiffness requirements of stile structure are found in the group of short stiles. It can also be noted that an increase in the height of test elements (from 80 mm by half) should result in a significant improvement in the structure stiffness reducing the deflection arrow by more than 75% for long stiles and about 55% for short stiles. Additional stress modeling analyses have shown that, using the same raw material for both solid and glued stiles, the abandonment of using solid timber in favor of glued timber constructed from friezes of 200 mm or 400 mm in length, does not result in a loss of stiffness of such elements, and in some cases results in a slight improvement in stiffness by reducing the deflection arrow of the element during normative loading. A comparison of the strength and stiffness of solid and glued stiles using simulated construction of models made of only one raw material (pine, country of origin: Poland, strength class C24/KS) is presented in Table 11, Table 12, Table 13, Table 14, Table 15 and Table 16.

The strength of any structure, including the tested stiles, is directly related to the value and distribution of normal stresses generated in the material under load. The analyses carried out prove that in none of the analyzed cases—taking into account the nature of the load, the geometry of the elements, and also the type of raw material used and its modification—is there any danger of exceeding the value of bending wood strength and thus destroying the component. As evidenced by other studies [66,67,68,69], the authors of numerical analyses most often conduct calculations not for individual elements, but for larger subassemblies. This study allows for the assumption that stiles comprising an element of a larger subassembly (e.g., a bed upholstery frame), through the constraints of freedom imposed by means of bonding with other elements have a reduced deflection character and thus generate much lower values of deflection and normal stresses. An example of this is the bed upholstery frame, which despite the very high loads from the tension force of wave springs (160–210 N for one spring) as a subassembly, is very rigid and strong [67]. A graphical summary of the deflection values of individual stiles depending on the version (A–F), as well as the origin of wood and the strength class of material is shown in Figure 8.

The results obtained in the modeling process indicate a deviation from the values obtained in the laboratory testing process. In relation to the values of stresses confirmed for each group of samples depending on the habitat of obtaining raw material, it was indicated that there is no relationship between the origin of the raw material and the maximum stress limits obtained. In the case of the tested raw material for normative loads, the influence of the origin of the raw material on the deformation susceptibility of the elements was confirmed. The results of deformation modeling coincide with the values of real tests.

## 4. Conclusions

The tests confirmed that the highest stresses are generated at high deflections of a given element. It can be observed that with the assumed type of normal loads in the long frames (in the A–C version of the elements) the stresses amounted to about 30 MPa and are higher than the stresses generated in frames of the same length, from the same raw materials, but of greater height. From the analyses carried out, basic conclusions were made about the value of normal stresses in the elements:There is no direct influence in the range of normal stresses on the modeling results from the type of coniferous raw material proposed,The influence of the form of modification of the element (solid-glued) on the modeling results was not confirmed,The stress level is lower when using elements with higher cross-sectional dimensions,The stress level is lower for shorter elements,Normative stresses do not significantly threaten the strength of the tested elements when the quality requirements of the material used are met.

Modeling due to stiffness and strength of individual elements is limited by the bed frame design. In the use of wood of Polish or Scandinavian pine species with a minimum strength class of 30 N/mm^2^, it is required to separate elements that meet the requirements of KW (C30) classes. This assumption translates into the need for rigorous sorting of lumber intended for processing [70,71,72,73]. An element conducive to the reduction of defects are optimization processes that result in increased strength ratings of semi-finished products directed to be joined into furniture stiles.

## Figures and Tables

**Figure 1 materials-15-06084-f001:**
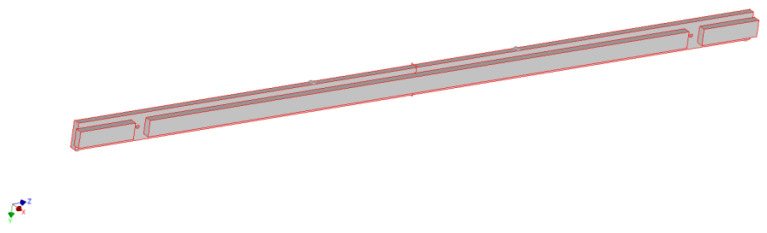
Selected geometric model of the continuous long frame version A made of solid wood material.

**Figure 2 materials-15-06084-f002:**
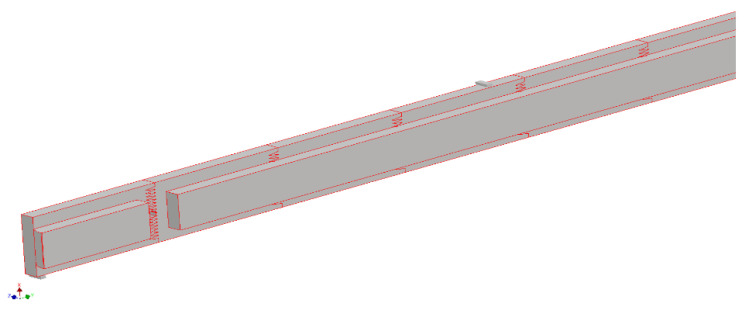
Fragment of the selected geometric model of the continuous long frame version B made of glued material with a finger-jointed lamella length of 200 mm.

**Figure 3 materials-15-06084-f003:**
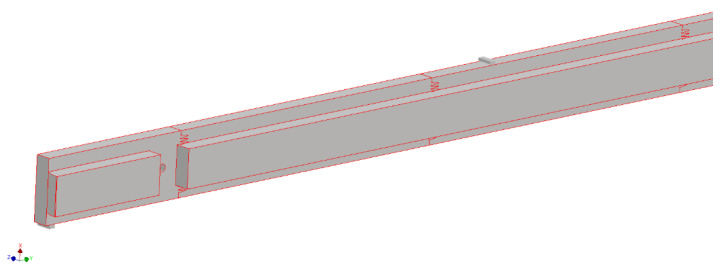
Fragment of a selected geometric model of a continuous C-version long frame made of glued material with a finger-jointed lamella length of 400 mm.

**Figure 4 materials-15-06084-f004:**
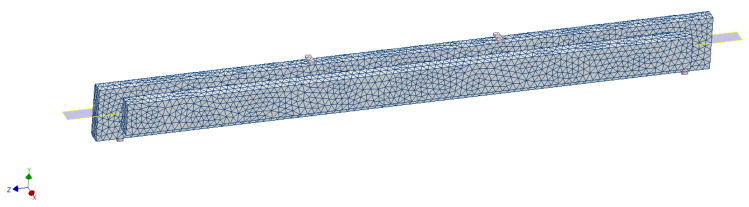
Selected mesh (computational) model of the D version of the short frame made of solid wood material.

**Figure 5 materials-15-06084-f005:**
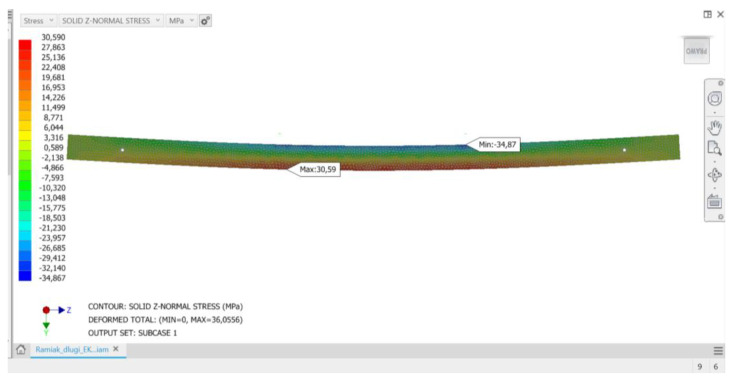
Mesh (computational) model of A-frame made of solid wood material with distribution of normal stress map.

**Figure 6 materials-15-06084-f006:**
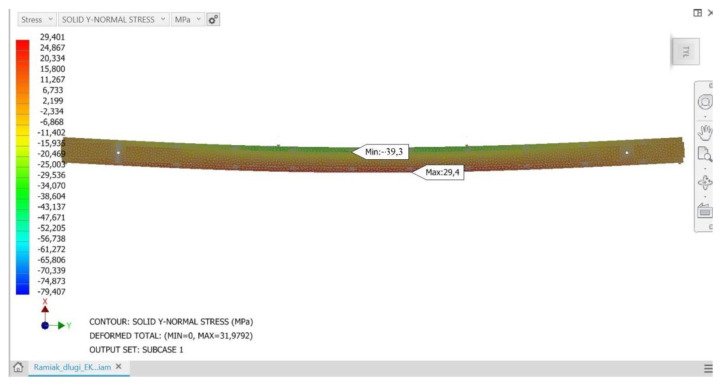
Mesh (computational) model of a version B frame made of glued material with a finger-jointed lamella length of 200 mm, together with the distribution of the normal stress map.

**Figure 7 materials-15-06084-f007:**
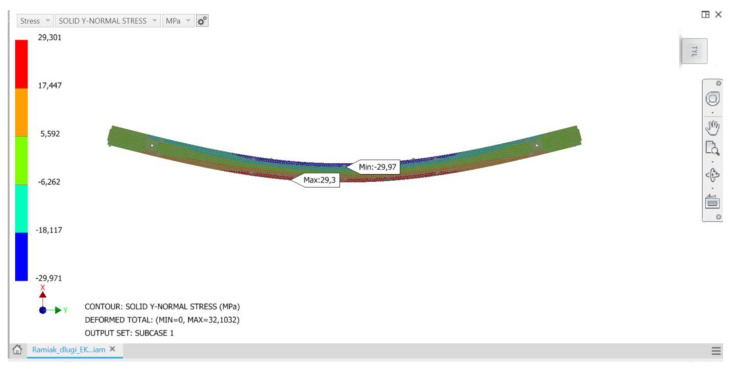
Mesh (computational) model of a C-version long frame made of glued material with a finger-jointed lamella length of 400 mm, along with the distribution of the normal stress map.

**Figure 8 materials-15-06084-f008:**
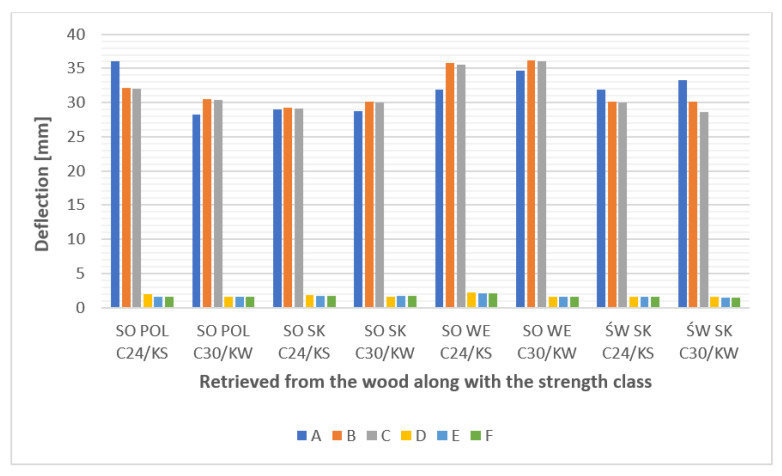
Graphical summary of the deflection values of individual frames depending on the version (A–F) and the origin of the wood and the strength class of the material.

**Table 1 materials-15-06084-t001:** Material data used in the analysis of numerical models of long frames in version A made of solid wood material.

Wood Species	Retrieved from	Density [kg/m^3^]		MOE [N/mm^2^]	
		C24	C30	C24	C30
PNSY	PL	468	517	8100	10,300
	EW	477	453	9100	8400
	SC	500	504	10,000	10,100
PCAB	SC	439	433	9100	8700

**Table 2 materials-15-06084-t002:** Material data used in the analysis of numerical models of B version long frames made of glued material.

Wood Species	Retrieved from	Density [kg/m^3^]		MOE [N/mm^2^]	
		C24	C30	C24	C30
PNSY	PL	440	458	9000	9500
	EW	499	473	8100	8000
	SC	491	504	9900	9600
PCAB	SC	443	431	9600	10,100

**Table 3 materials-15-06084-t003:** Material data used in the analysis of numerical models of D version short frames made of solid wood material.

Wood Species	Retrieved from	Density [kg/m^3^]		MOE [N/mm^2^]	
		C24	C30	C24	C30
PNSY	PL	501	524	10,600	12,500
	EW	425	419	9100	12,900
	SC	430	448	11,400	12,800
PCAB	SC	434	435	12,700	13,200

**Table 4 materials-15-06084-t004:** Material data used in the analysis of numerical models of E–F version short frames made of glued material.

Wood Species	Retrieved from	Density [kg/m^3^]		MOE [N/mm^2^]	
		C24	C30	C24	C30
PNSY	PL	524	453	12,500	12,300
	EW	402	415	9600	12,600
	SC	441	454	11,700	11,900
PCAB	SC	434	429	12,300	13,300

**Table 5 materials-15-06084-t005:** Normal stresses and deflection (displacement) of the A version long frames made of solid wood material.

Wood Species	Retrieved from	Strength Class	Normal Stress [N/mm^2^]	Deflection [mm]
PNSY	PL	C24/KS	30.6	36.0
		C30/KW		28.2
	SC	C24/KS		29.0
		C30/KW		28.7
	EW	C24/KS		31.9
		C30/KW		34.7
PCAB	SC	C24/KS		31.9
		C30/KW		33.3

**Table 6 materials-15-06084-t006:** Normal stresses and deflection (displacement) of the B version long frames made of glued material with a finger-jointed lamella length of 200 mm.

Wood Species	Retrieved from	Strength Class	Normal Stress [N/mm^2^]	Deflection [mm]
PNSY	PL	C24/KS	29.4	32.2
		C30/KW	29.5	30.5
	SC	C24/KS		29.3
		C30/KW		30.2
	EW	C24/KS		35.8
		C30/KW		36.2
PCAB	SC	C24/KS		30.2
		C30/KW		30.2

**Table 7 materials-15-06084-t007:** Normal stresses and deflection (displacement) of C-version long frames made of glued material with a finger-jointed lamella length of 400 mm.

Wood Species	Retrieved from	Strength Class	Normal Stress [N/mm^2^]	Deflection [mm]
PNSY	PL	C24/KS	29.3	32.0
		C30/KW		30.4
	SC	C24/KS		29.1
		C30/KW		30.0
	EW	C24/KS		35.6
		C30/KW		36.0
PCAB	SC	C24/KS		30.0
		C30/KW		28.6

**Table 8 materials-15-06084-t008:** Normal stresses and deflection (displacement) of D version short frames made of solid wood material.

Wood Species	Retrieved from	Strength Class	Normal Stress [N/mm^2^]	Deflection [mm]
PNSY	PL	C24/KS	12.1	1.9
		C30/KW		1.6
	SC	C24/KS		1.8
		C30/KW		1.6
	EW	C24/KS		2.2
		C30/KW		1.6
PCAB	SC	C24/KS		1.6
		C30/KW		1.6

**Table 9 materials-15-06084-t009:** Normal stresses and deflection (displacement) of E version short frames made of glued material with a finger-jointed lamella length of 200 mm.

Wood Species	Retrieved from	Strength Class	Normal Stress [N/mm^2^]	Deflection [mm]
PNSY	PL	C24/KS	11.5	1.6
		C30/KW		1.6
	SC	C24/KS		1.7
		C30/KW		1.7
	EW	C24/KS		2.1
		C30/KW		1.6
PCAB	SC	C24/KS		1.6
		C30/KW		1.5

**Table 10 materials-15-06084-t010:** Normal stresses and deflection (displacement) of F version short frames made of glued material with a finger-jointed lamella length of 400 mm.

Wood Species	Retrieved from	Strength Class	Normal Stress [N/mm^2^]	Deflection [mm]
PNSY	PL	C24/KS	11.0	1.6
		C30/KW		1.6
	SC	C24/KS		1.7
		C30/KW		1.7
	EW	C24/KS		2.1
		C30/KW		1.6
PCAB	SC	C24/KS		1.6
		C30/KW		1.5

**Table 11 materials-15-06084-t011:** Normal stresses and deflection (displacement) of long frames in version A made of solid wood material (pine, country of origin: Poland).

Wood Species	Retrieved from	Strength Class	Normal Stress [N/mm^2^]	Deflection [mm]
PNSY Sosna	Polska PL	C24/KS	30.59	36.0
		C30/KW	28.8	28.2

**Table 12 materials-15-06084-t012:** Normal stresses and deflection (displacement) of long frames in version B made of glued material with a finger-jointed lamella length of 200 mm (pine, country of origin: Poland).

Wood Species	Retrieved from	Strength Class	Normal Stress [N/mm^2^]	Deflection [mm]
PNSY	PL	C24/KS	29.5	35.8
		C30/KW	29.5	28.2

**Table 13 materials-15-06084-t013:** Normal stresses and deflection (displacement) of C-version long frames made of glued material with a finger-jointed lamella length of 400 mm (pine, country of origin: Poland).

Wood Species	Retrieved from	Strength Class	Normal Stress [N/mm^2^]	Deflection [mm]
PNSY	PL	C24/KS	29.3	35.6
		C30/KW	29.3	28.0

**Table 14 materials-15-06084-t014:** Normal stresses and deflection (displacement) of D version short frames made of solid wood material (pine, country of origin: Poland).

Wood Species	Retrieved from	Strength Class	Normal Stress [N/mm^2^]	Deflection [mm]
PNSY	PL	C24/KS	12.1	1.9
		C30/KW	12.1	1.6

**Table 15 materials-15-06084-t015:** Normal stresses and deflection (displacement) of E version short frames made of glued material with a finger-jointed lamella length of 200 mm (pine, country of origin: Poland).

Wood Species	Retrieved from	Strength Class	Normal Stress [N/mm^2^]	Deflection [mm]
PNSY	PL	C24/KS	11.5	1.9
		C30/KW	11.5	1.6

**Table 16 materials-15-06084-t016:** Normal stresses and deflection (displacement) of F version short frames made of glued material with a finger-jointed lamella length of 400 mm (pine, country of origin: Poland).

Wood Species	Retrieved from	Strength Class	Normal Stress [N/mm^2^]	Deflection [mm]
PNSY	PL	C24/KS	11.0	1.9
		C30/KW	11.0	1.6

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
