# Peer review of "Elements of Designing Upholstered Furniture Sandwich Frames Using Finite Element Method"

_materials, 2022, doi:10.3390/ma15176084_

Round 1
Reviewer 1 Report
General analysis:
* The use of a very well known tool as FEM analysis using a normal or well stablished methodology to a wooden loadbearing member can not be considered by itself as an innovation in research, or a research. Its use applied to furniture analysis is similar to the use of this tool applied to structural elements, for which it has been extensively applied. So although using FEM analysis in furniture can be technically interesting for design purposes it is not adding new scientific knowledge.
* The use of finger-jointed elements in wood framing products is stablished since long time ago, as it is a fundamental part of the manufacturing of many stablished products such as Glued Laminated Timber. It is already known that the glued finger-joint of several solid wood sections to form a lamella is equivalent or better in stiffness and strength to the solid wood that is being used (in fact the finger-jointing is used to improve the overall quality of the wood used for many product, by removing defects such as knots, etc.). For the former reasons the analysis of the use of finger jointed material is not adding new scientific knowledge (and also its effect has not been specifically assessed using the FEM model).
* No relevant scientific conclusions are obtained. As mostly:
- the results are related to the modulus of elasticity used, and that is something obvious and well known.
- the conclusions are operational results of a technical analysis and not scientific, since no new knowledge is coming from the analysis (it is already known that grading the wood is a good practice and the results are somehow trivial as as explained before the finger-jointing has been thoroughly assessed in the past).
* Flaws in what regards to scientific method application are present:
- In many cases the references are in excess and improperly used because they are not related to the information discussed, for example reference 6 is cited and it does not relate to any analysis (is related to leave decomposition that has not incidence in the paper contents) or reference 10 is related to thermally treated wood that is not mentioned in the research.
- There is not a proper description of the test analysis performed and its aim: the number of samples and their dimensions are not indicated, nor it is the standard procedure used, the results do not indicate the variability, if they are average values, etc.
- Despite that the methods for jointing the pieces are repeatedly addressed then they are not used in the FEM modelling, or if they were used are not explained or described in the text, nor their results assessed. No specific adhesive or other joint method is specifically assessed in what has been seen in the paper.
- The dimensions of the specimens assessed are not declared or indicated, which is a serious flaw as it would not allow the repetition of the analysis by other scientists.
* Problems were observed in the wood mechanical properties analysis:
- Since the structural grades system is well stablished in Europe, with significant standards such as EN 1912 stating the strength classes corresponding to the different grades for each country (based in national grading standards) it is unnecessary to test many of the woods to obtain their properties (for example picea abies coming from Scandinavia).
- It can be admitted that for a more detailed information the authors preferred to test, but the results obtained in the work in many cases are not matching the current values officially stated for the strength classes since for example in table 1 the MOE values for Poland and for the other origins are inferior to those stated officially for a C24 strength class (and therefore they would not correspond to this strength class). Is the test method used according to EN standard EN 408?..
- Testing short members of wood can lead to a increased effect of the shear deformation which maybe can explain the lower MOE values of the short pieces, but is not a suitable way of testing for general mechanical properties characterization.
Other issues in the text:
* The title include the phrase “Sandwich frame” and this type of structure is missing according to the classical definition of sandwich element (two faces glued to a core).
* Differences between what it is stated in the abstract in lines 16-20 (the deformation and strength behavior was improved.., stapled corner joints..).. in the results and materials only a general type of section was included (1 main piece and other non continuous adjoined in one face), no staples or reinforcement was discussed.
* Figure 6 must be reviewed or explained
* English must be reviewed (mainly in technical vocabulary, for example Jigsaw would not be correct, maybe finger-jointed lamella sections would be better, and in some occasions the explanations are not clear, in other occassions are fine).
Author Response
Reviewer 1
Thank You very much for your thorough evaluation of our publication. Your comments and corrections are very valuable. They represent a significant improvement in the quality of the publication. We hope that the present explanations will be satisfactory to You.
With best regards
Authors
The problem for the paper:
Reviewer: * The use of finger-jointed elements in wood framing products is stablished since long time ago, as it is a fundamental part of the manufacturing of many stablished products such as Glued Laminated Timber. It is already known that the glued finger-joint of several solid wood sections to form a lamella is equivalent or better in stiffness and strength to the solid wood that is being used (in fact the finger-jointing is used to improve the overall quality of the wood used for many product, by removing defects such as knots, etc.). For the former reasons the analysis of the use of finger jointed material is not adding new scientific knowledge (and also its effect has not been specifically assessed using the FEM model).
Answer: The variability of elements joined in length and width (two layers) is part of the verification of changes in the use of wood for furniture. The Reviewer rightly emphasizes the use of FEM for structures, but in furniture, this form may have additional significance due to the functional characteristics of the furniture and not just strength.
Reviewer: * No relevant scientific conclusions are obtained. As mostly:
- the results are related to the modulus of elasticity used, and that is something obvious and well known.
Answer: As the Reviewer rightly points out, the modulus of elasticity is the primary indicator used in FEM. However, for our study, reference was made to actual laboratory-confirmed results to indicate that the index is relative and dependent on the batch of wood raw material.
Reviewer: - the conclusions are operational results of a technical analysis and not scientific, since no new knowledge is coming from the analysis (it is already known that grading the wood is a good practice and the results are somehow trivial as as explained before the finger-jointing has been thoroughly assessed in the past).
Answer: The conclusions have been corrected. The Reviewer is, of course, right, which has been taken into account. At the same time, the practice of sorting requires specifying the features of classification and adopting rules for certain defects or quality of raw material. Therefore, the authors presented results indicating the factor of application of normative loads have a decisive role in reducing the influence of other factors of raw material type.
* Flaws in what regards to scientific method application are present:
Reviewer: - In many cases the references are in excess and improperly used because they are not related to the information discussed, for example reference 6 is cited and it does not relate to any analysis (is related to leave decomposition that has not incidence in the paper contents) or reference 10 is related to thermally treated wood that is not mentioned in the research.
Answer: A valid comment. The authors made too broad a reference to the influence of other growth and use factors on changes in shaping the quality of wood headed for furniture processing.
Reviewer: - There is not a proper description of the test analysis performed and its aim: the number of samples and their dimensions are not indicated, nor it is the standard procedure used, the results do not indicate the variability, if they are average values, etc.
Answer: It was supplemented by providing the number of laboratory samples indicating the number of average values from the results obtained.
Reviewer: - Despite that the methods for jointing the pieces are repeatedly addressed then they are not used in the FEM modelling, or if they were used are not explained or described in the text, nor their results assessed. No specific adhesive or other joint method is specifically assessed in what has been seen in the paper.
Answer: The reviewer's remark is correct. The information on the use of bonded raw material in the description has been completed
Reviewer: - The dimensions of the specimens assessed are not declared or indicated, which is a serious flaw as it would not allow the repetition of the analysis by other scientists.
Answer: The dimensions of the samples are given in the figures but, as the reviewer rightly notes, have been supplemented in the text.
Reviewer: * Problems were observed in the wood mechanical properties analysis:
- Since the structural grades system is well stablished in Europe, with significant standards such as EN 1912 stating the strength classes corresponding to the different grades for each country (based in national grading standards) it is unnecessary to test many of the woods to obtain their properties (for example picea abies coming from Scandinavia).
Answer: A good point but as the Reviewer certainly acknowledges, raw timber is constantly changing the influence of habitat plays a significant role in shaping quality. As a specialist, the Reviewer is aware that standards are generalizations of quality characteristics and are mostly supporting documents for calculations. Quality characteristics can significantly deviate from the norm data (for example, the effect of quantum on the proportion of elements that do not reach the assumed strength parameters). Hence the need for research to solidify the knowledge of errors in standards and the need to revise them.
Reviewer: - It can be admitted that for a more detailed information the authors preferred to test, but the results obtained in the work in many cases are not matching the current values officially stated for the strength classes since for example in table 1 the MOE values for Poland and for the other origins are inferior to those stated officially for a C24 strength class (and therefore they would not correspond to this strength class). Is the test method used according to EN standard EN 408?..
Answer: The Reviewer has rightly made a comment. However, the Currently reported values for materials are a generalization for wood raw material. Therefore, the study was based on laboratory results reflecting the actual MOE index for the raw material under study. This indicates the high variability of raw material strength parameters which should prompt researchers to carefully select data when modeling. The reviewer certainly agrees with the necessity of verifying previous research results for deeper knowledge of their variability.
Reviewer: - Testing short members of wood can lead to a increased effect of the shear deformation which maybe can explain the lower MOE values of the short pieces, but is not a suitable way of testing for general mechanical properties characterization.
Answer: Note right. The Reviewer confirms the distribution of stresses in tests loaded with stress components, Tests for short elements are obviously intended to indicate the trend of stress distribution, in FEM it is necessary to make a broader reference to the distribution of forces in the tested with respect to real conditions.
Other issues in the text:
Reviewer: * The title include the phrase “Sandwich frame” and this type of structure is missing according to the classical definition of sandwich element (two faces glued to a core).
Answer: Clarification on the use of glulam has been completed
Reviewer: * Differences between what it is stated in the abstract in lines 16-20 (the deformation and strength behavior was improved.., stapled corner joints..).. in the results and materials only a general type of section was included (1 main piece and other non continuous adjoined in one face), no staples or reinforcement was discussed.
Answer: Comment justified . Supplemented the article with an explanation of the structure of glulam. At the same time, we clarify that the staples acted as pre-stabilizing elements for the lamellas during the glue bonding process.
Reviewer: * Figure 6 must be reviewed or explained
Answer: Figure corrected
Reviewer: * English must be reviewed (mainly in technical vocabulary, for example Jigsaw would not be correct, maybe finger-jointed lamella sections would be better, and in some occasions the explanations are not clear, in other occassions are fine).
Answer: Language corrected
The review is very insightful and, thanks to the reviewer's comments, significantly raises the level of quality of the prepared article. For all comments we thank you very much
We thank the Reviewer for important comments that enhance the work.
Reviewer 2 Report
In the review of the manuscript titled: Elements of designing upholstered furniture sandwich frames using finite element method. The authors have provided a good description and the methodology is also fine. I would like to see this review publish but after some questions as follow;
1. Why did the authors just focus on the finite element method?
2. What is the effect of orthotropic properties of solid pine wood on furniture frames?
3. How did the authors obtain displacement and equivalent strain distributions?
4. The authors are requested to compare the results obtained with the FEM method with previously reported methods.
5. The authors used CAE Autodesk Inventor Nastran finite element method; linear static analyses were performed by simulating normative loading. The authors are requested to provide some discussion about the simulation.
Author Response
Reviewer 2
Thank You very much for your thorough evaluation of our publication. Your comments and corrections are very valuable. They represent a significant improvement in the quality of the publication. We hope that the present explanations will be satisfactory to You.
With best regards
Authors
The problem for the paper.
Reviewer: 1. Why did the authors just focus on the finite element method?
Answer: Presentation of the principles of furniture design as wooden components of furniture construction, with classification and their characteristics, is concerned with taking into account the requirements of safety in use. Basic strength testing methods can be prone to design errors, characterization of materials, elements and structures. The FEM method addresses the issue of calculating and verifying the stiffness and strength of components, joints and entire structures, including issues of user health hazards. The use of modeling for the evaluation of furniture structures is to serve as one of the primary sources of knowledge for the formulation of design assumptions for a new product, often reveals deviations from the general assumptions of structural design, which significantly improves the process of finding solutions that satisfy the creators of a new piece of furniture. The FEM method is a developmental one for the direction of designing furniture as complex structures. The FEM method is an engineering tool to help solve technical problems in furniture design.
Reviewer: 2. What is the effect of orthotropic properties of solid pine wood on furniture frames?
Answer: As an orotropic material, wood has variable mechanical properties depending on the considered direction of loads. In engineering terms, this means that it is necessary to take into account the directions of forces s in relation to the direction of the wood fibers. It transmits stresses along the fibers much better than across them. The use of longitudinal and lengthwise glulam, on the one hand, reduces the influence of the orotropic features of the tangential and transverse section, but does not eliminate the influence of the longitudinal section. However, for the purpose of FEM modeling of force distribution in the normative loading process, the reference to the stresses acting in the longitudinal structure is taken into account primarily. This allows a partial simplification of the model of the glulam component under study.
Reviewer: 3. How did the authors obtain displacement and equivalent strain distributions?
Answer: The distribution of reduced stresses as well as the distribution of deformations was obtained by using discrete models and an advanced iterative method defined in the Autodesk Nastran solver (processor). There are two ways to arrive at the discrete model and one of them, used during the analysis in this paper, is FEA - the finite element method. The FEA method is an approximate method. It is also a numerical tool designed to solve differential equations, and more specifically to find detailed solutions. The discretization of the structure of each frame consisted of automatically dividing all model elements into appropriately selected finite elements of defined shape and properties, connected to each other at specific points called nodes. All models were automatically divided into nonlinear finite elements (parabolic type) with an assumed element size of 10 mm. The models were loaded exactly as they were in experimental tests on real elements. All this - after the calculations - thanks to the work of the so-called post-processor, it was possible to automatically visualize the distribution of stresses and equivalent strains.
Reviewer: 4. The authors are requested to compare the results obtained with the FEM method with previously reported methods.
Answer: The results obtained in the modeling process indicate a deviation from the values obtained in the laboratory testing process. In relation to the values of stresses confirmed for each group of samples depending on the habitat of obtaining raw material, it was indicated that there is no relationship between the origin of the raw material and the maximum stress limits obtained. In the case of the tested raw material for normative loads, the influence of the origin of the raw material on the deformation susceptibility of the elements was confirmed. The results of deformation modeling coincide with the values of real tests.
Reviewer: 5. The authors used CAE Autodesk Inventor Nastran finite element method; linear static analyses were performed by simulating normative loading. The authors are requested to provide some discussion about the simulation
Answer: Analysis of stiffness and strength of selected frame structures, in the form of 3D models, was carried out virtually in Autodesk Inventor Nastran version 2020 using the finite element method (FEM). Prior to working in Autodesk Inventor Nastran, continuous geometric models, i.e. 3D models of the frame structures, were developed on the basis of engineering sketches - using Autodesk Inventor. Due to the need to realistically reflect the experimental (laboratory) tests carried out in parallel, supports and thrusters were introduced into each 3D model to simulate the four-point bending test. Then the so-called idealization was carried out, which consisted of assigning specific physical properties of the raw material (material) to the individual elements of the frames. The physical properties of each type of raw material were derived from parallel experimental (laboratory) tests. The next step was to automatically divide all model elements into appropriately selected finite elements of defined shape and properties, define boundary conditions in the form of determining the number and type of degrees of freedom at selected nodes, assigning interactions (contacts) between selected surfaces, and introducing external loading of the structure. After preparing the models for calculation, numerical analyses were carried out in the linear static analysis mode. It should be noted that the authors made a number of tests and comparisons of the results obtained through experimental studies and numerical analyses before conducting the actual numerical analyses. The obtained data indicated differences of 1-3%, which should be considered a very good convergence of results and thus correct preparation of models for numerical calculations.
We thank the Reviewer for important comments that enhance the work.
Reviewer 3 Report
Research on optimizing the design of seating or bed furniture is always welcome. These structures are always challenging to optimize and obtain higher strength and deformation characteristics. Therefore the present study is relevant for publication.
I have the following remarks and suggestions to the authors:
· They are clearly seen from the drawings that there are finger joints on some of the test samples. Unfortunately, it is no anywhere an explanation for this.
· Row 96-98 – the sentence "The study of wooden chair frames using the was undertaken by researchers [50] using the stress-strain methodology developed by Marinov [51] for FEM-based analysis of box furniture structures." have to be corrected; there are missing words in it.
· Row 98 - Marinov [51] has to be corrected as Marinova [51].
· Row 110 - Analizę sztywności oraz wytrzymałości wybranych konstrukcji ramiaków w postaci - To be translated!
· Tables 1 to 4 – the C24 and C30 are not explained, it is mention PN-EN 338, but it is not defined that “C” correspond to “KW”
· In the whole article I recommend “Solid material” to be changed with “Solid wood material”
Generally, the conclusions are not clear enough. It is recommended to be improved.
Author Response
Reviewer 3
Thank You very much for your thorough evaluation of our publication. Your comments and corrections are very valuable. They represent a significant improvement in the quality of the publication. We hope that the present explanations will be satisfactory to You.
With best regards
Authors
Reviewer: They are clearly seen from the drawings that there are finger joints on some of the test samples. Unfortunately, it is no anywhere an explanation for this.
Answer: As an orotropic material, wood has variable mechanical properties depending on the considered direction of loads. In engineering terms, this means that it is necessary to take into account the directions of forces s in relation to the direction of the wood fibers. It transmits stresses along the fibers much better than across them. The use of wood glued longitudinally and lengthwise with a finger joint (a 13mm finger-length furniture joint using PVAC glue) ) and on the one hand reduces the influence of the orotropic features of the tangential and transverse sections but does not eliminate the influence of the longitudinal section. However, for the purpose of FEM modeling of force distribution in the normative loading process, the reference to the stresses acting in the longitudinal structure is mainly considered. This allows a partial simplification of the model of the glulam component under study.
Reviewer: · Row 96-98 – the sentence "The study of wooden chair frames using the was undertaken by researchers [50] using the stress-strain methodology developed by Marinov [51] for FEM-based analysis of box furniture structures." have to be corrected; there are missing words in it.
Answer: Corrected
Reviewer: · Row 98 - Marinov [51] has to be corrected as Marinova [51].
Answer: Corrected
Reviewer: · Row 110 - Analizę sztywności oraz wytrzymałości wybranych konstrukcji ramiaków w postaci - To be translated!
Answer: Corrected
Reviewer: ·Tables 1 to 4 – the C24 and C30 are not explained, it is mention PN-EN 338, but it is not defined that “C” correspond to “KW
Answer: Corrected
Reviewer: In the whole article I recommend “Solid material” to be changed with “Solid wood material”
Answer: Corrected
We thank the Reviewer for important comments that enhance the work.
Round 2
Reviewer 1 Report
Thank you for your comments and clarifications. I see that some explanations have improved although not others, for example the use of excessive references not related to the paper objective is still visible, and also the test procedure used (standard) is not included, etc.
In my opinion the main drawback of the work for being published in a scientific journal is that it is mainly a technical analysis, rather than a research paper with new non previously available information.
Author Response
Reviewer 1
Thank You very much for your thorough evaluation of our publication. Your comments and corrections are very valuable. They represent a significant improvement in the quality of the publication. We hope that the present explanations will be satisfactory to You.
With best regards
Authors
The problem for the paper:
Reviewer: I see that some explanations have improved although not others, for example the use of excessive references not related to the paper objective is still visible,
Answer: Corrected
Removed inappropriate elements of the literature review
Reviewer: * and also the test procedure used (standard) is not included, etc
Answer: Corrected
The test method used for evaluating laboratory results and output for subsequent modeling is in accordance with EN 408.
The system of strength classes is based in Europe on the provisions of standards, such as EN 1912 or EN 338. In the study, it was assumed that, despite the existence of differences in the tested parameters for machine strength testing and visual evaluation 94021 of solid elements, it is possible to take into account the results of laboratory self-testing d modeling of the construction of glued frames.
EN 408:2010+A1:2012 Wood structures -- Structural solid and glued laminated timber -- Determination of certain physical and mechanical properties
EN 1912:2012 Structural timber -- Strength classes -- Visual division into classes and grades
We thank the Reviewer for a very thorough review of the paper. Thanks to the Reviewer's comments, we significantly improve the quality level of the prepared article. For all the comments we again thank you very much
We would like to thank the Reviewer for his comments, which improve the quality of the paper.